# Hydroxynonenal causes Langerhans cell degeneration in the pancreas of Japanese macaque monkeys

**Piyakarn Boontem[1], Tetsumori Yamashima[1,2]***

**1** Departments of Cell Metabolism and Nutrition, Kanazawa, Japan, **2** Psychiatry and Behavioral Science, Kanazawa, Japan

* yamashima215@gmail.com

## Abstract

### Background

For their functions of insulin biosynthesis and glucose- and fatty acid- mediated insulin secretion, Langerhans β-cells require an intracellular milieu rich in oxygen. This requirement makes β-cells, with their constitutively low antioxidative defense, susceptible to the oxidative stress. Although much progress has been made in identifying its molecular basis in experimental systems, whether the oxidative stress due to excessive fatty acids plays a crucial role in the Langerhans cell degeneration in primates is still debated.

### Methods

Focusing on Hsp70.1, which has dual functions as molecular chaperone and lysosomal stabilizer, the mechanism of lipotoxicity to Langerhans cells was studied using macaque monkeys after the consecutive injections of the lipid peroxidation product 'hydroxynonenal'. Based on the '*calpain-cathepsin hypothesis*' formulated in 1998, calpain activation, Hsp70.1 cleavage, and lysosomal integrity were studied by immunofluorescence histochemistry, electron microscopy, and Western blotting.

### Results

Light microscopy showed more abundant vacuole formation in the hydroxynonenal-treated islet cells than the control cells. Electron microscopy showed that vacuolar changes, which were identified as enlarged rough ER, occurred mainly in β-cells followed by δ-cells. Intriguingly, both cell types showed a marked decrease in insulin and somatostatin granules. Furthermore, they exhibited marked increases in peroxisomes, autophagosomes/ autolysosomes, lysosomal and peroxisomal membrane rupture/permeabilization, and mitochondrial degeneration. Disrupted peroxisomes were often localized in the close vicinity of degenerating mitochondria or autolysosomes. Immunofluorescence histochemical analysis showed an increased co-localization of activated μ-calpain and Hsp70.1 with the extralysosomal release of cathepsin B. Western blotting showed increases in μ-calpain activation, Hsp70.1 cleavage, and expression of the hydroxynonenal receptor GPR109A.

**Data Availability Statement:** All relevant data are within the paper and its Supporting Information files.

**Funding:** This work was supported by a grant from Kiban-Kenkyu (B) (19H04029) from the Japanese

Ministry of Education, Culture, Sports, Science and Technology. The funders had no role in study design, data collection and analysis, decision to publish, or preparation of the manuscript.

**Competing interests:** The authors declare that they have no conflict of interest.

**Abbreviations:** ER, endoplasmic reticulum; Hsp70.1, heat-shock protein 70.1; $H_2O_2$, hydrogen peroxide; OH•, hydroxyl radicals; NEFAs, nonesterified fatty acids; ROS, reactive oxygen species; O2•–, superoxide radicals.

## Conclusions

Taken together, these data implicate hydroxynonenal in both oxidation of Hsp70.1 and activation of μ-calpain. The calpain-mediated cleavage of the carbonylated Hsp70.1, may cause lysosomal membrane rupture/permeabilization. The low defense of primate Langerhans cells against hydroxynonenal and peroxisomally-generated hydrogen peroxide, was presumably overwhelmed to facilitate cell degeneration.

## Introduction

The intracellular milieu of pancreatic β-cells is rich in oxygen, glucose and fatty acids for insulin biosynthesis and secretion. This makes β-cells, with their constitutively low enzymatic antioxidative defense equipment [1, 2], susceptible to oxidative stress during glucolipotoxicity. However, the mechanism of β-cell degeneration and death is not fully understood in type 2 diabetes until now. Hyperlipidemia (lipotoxicity) is accepted to be the greatest risk factor [3], because non-esterified fatty acids (NEFAs) are potent inducers of reactive oxygen species (ROS) through different mechanisms. β-cells exposed to oxidative, excessive or conjugated NEFAs develop cell degeneration/death [3–5]. The molecular mechanisms underlying the oxidative injury due to ROS, comprise of endoplasmic reticulum (ER) stress, mitochondrial dysfunction, impaired autophagy, lysosomal disintegrity, inflammation, and mixture of them [6–11].

These molecular events are grossly similar to the neurodegeneration of hypothalamic arcuate nucleus occurring after the intake of high-fat diets [12] or hippocampal CA1 sector exposed to the oxidative stress during reperfusion after transient ischemia [3, 13]. In 1998, Yamashima and his colleagues formulated the 'calpain-cathepsin hypothesis' as a mechanism of ischemic neuronal death [14, 15]. Thereafter, they suggested role of lysosomal rupture due to calpain-mediated cleavage of oxidized (carbonylated) heat-shock protein 70.1 (Hsp70.1; human type of Hsp70, also called Hsp72) as a mechanism of neuronal death [5, 16–18]. However, previous researchers have not found so far any convincing molecular mechanism of a major contribution of ROS to the β-cell degeneration and death. Antioxidative defence mechanisms of β-cells would be overwhelmed by overproduction of ROS [19] such as superoxide radicals (O2•–), hydrogen peroxide ($H_2O_2$), and the most reactive and toxic hydroxyl radicals (OH•). Among these, OH• is responsible for the carbonylation of Hsp70.1 [16, 17], and its precursor $H_2O_2$ plays a central role in the deterioration of glucose tolerance in the development of type 2 diabetes [19]. The source of $H_2O_2$ in β-cells has been considered to be the electron transport chain in mitochondria. While this source is obviously important, peroxisomal generation of $H_2O_2$ is more crucial in β-cells [20–23].

Lysosomes and peroxisomes, being contained in all eukaryote cells, are similar at the ultrastructural level but hold distinct enzymes for the completely different function. Lysosomes were discovered by the Belgian cytologist Christian de Duve in the 1950s [24]. They contain a wide variety of hydrolytic enzymes (acid hydrolases such as cathepsins B, L, etc.) which were produced in the rough ER, and are responsible for the digestion of macromolecules, old cell parts, and microorganisms. In oxygen-poor areas with an acidic environment (~pH4.5), lysosomes break down macromolecules such as damaged/aged/misfolded proteins, nucleic acids, and polysaccharides for the recycling. For this reason, an ATP-driven proton pump is located inside the double membrane which borders the lysosomes in order to pump $H^+$ ions into their lumen. The double membrane surrounding the lysosome is vital to ensure cathepsins do not

leak out into the cytoplasm and damage the cell from within. Recent data advocate for dual roles of Hsp70.1 not only as a molecular chaperone for damaged proteins but also as a guardian of lysosomal membrane integrity [25, 26]. Therefore, in case of Hsp70.1 dysfunction, not only failure of protein traffic and degradation (autophagy failure) but also lysosomal destabilization (cell degeneration and death) may occur.

Peroxisomes are single membrane-bounded organelles that are best-known for their involvement in both the cellular lipid metabolism and the cellular redox balance [27–29]. Peroxisomes were first described by a Swedish doctoral student, J. Rhodin in 1954 [30], and were identified as a cell organelle by de Duve. He named them "peroxisomes", replacing the formerly used morphological term "microbodies" [31]. De Duve and Baudhuin discovered that peroxisomes contain several oxidases involved in the production of $H_2O_2$ as well as catalase involved in the decomposition of $H_2O_2$ to oxygen and water [31, 32]. So, 'peroxisomes' owe their name to 'hydrogen peroxide' -generating and -scavenging activities. Peroxisomes are small spheroid organelles with a fine, granular matrix. Peroxisomes hold diverse oxidative enzymes that were produced at free ribosomes and require oxygen. To absorb nutrients that the cell has acquired, peroxisomes digest long-chain fatty acids and break them down into smaller molecules by β-oxidation. One of the byproducts of the β-oxidation is $H_2O_2$, so peroxisomes break $H_2O_2$, down into water ($H_2O$) and $O_2$. $H_2O_2$ itself is not very reactive, but $Fe^{2+}$ or $Fe^{3+}$ reacts with $H_2O_2$ to produce OH• in mitochondria. Accordingly, in both health and diseases, peroxisomes and mitochondria have a close relationship each other. Peroxisome generation, maintenance, and turnover are of great importance for the cellular homeostasis and survival. Peroxisomes are degraded by themselves or when fusing with lysosomes. The degradation takes just ~4 min, and half-life of peroxisomes is only 5 days.

The pathogenesis of β-cell dysfunction leading to insulin resistance and impaired insulin secretion seen in type 2 diabetes is incompletely understood. Previous animal and in vitro studies demonstrated that chronic levels of oxidative stress are among the earliest abnormalities in the natural history of type 2 diabetes, although the evidence is insufficient [6]. On the other hand, the evidence for inflammation in the etiology of type 2 diabetes is broad, however, the underlying mechanism is also not well understood. Increased oxidative stress and inflammation, combined together, may lead to the pancreatic β-cell failure [1–5]. Elevated levels of inflammatory markers, pro-oxidants and various markers of oxidative tissue damage were found in the diabetic patients, which indicates their involvement in the occurrence of insulin resistance and impaired insulin secretion. Peroxidation of polyunsaturated fatty acids is intensified in cells subjected to oxidative stress and inflammation, and it results in the generation of various bioactive compounds [6–10]. Reactive aldehyde, especially hydroxynonenal, is considered as the representative cytotoxic product of lipid peroxidation causing long-lasting biological consequences, in particular, by covalent modification of macromolecules. Hydroxynonenal is a major aldehyde which was produced during the OH•-mediated peroxidation of ω-6 polyunsaturated fatty acids. In 2009, Oikawa et al. (2009) [16] found by the proteomic analysis that Hsp70.1 can be a target of oxidation (carbonylation) by hydroxynonenal. Subsequently, Sahara and Yamashima (2010) [33] demonstrated by the in-vitro experiment that Hsp70.1 being carbonylated by hydroxynonenal is prone to activated μ-calpain-mediated cleavage. Calpain-mediated cleavage of carbonylated Hsp70.1 leads to the lysosomal membrane rupture/permeabilization with the resultant release of cathepsins and cell death.

It is well-known that insulin resistance and impaired insulin signalling may be a contributory factor to the progression of type 2 diabetes. Although reduced release of insulin from β-cells is fundamental for both, the molecular mechanism of the Langerhans cell dysfunction caused by chronic inflammation and oxidative stress remains grossly unknown. Much progress has been made in identifying the molecular basis of lipotoxicity in the experimental

systems, however whether or not this phenomenon actually plays an important role for the Langerhans cell degeneration remains debated especially in the primates [10]. To overcome this controversy, the authors thought that the experimental paradigm using non-human primates is indispensable. In the preliminary study, we confirmed that the intravenous injections of the synthetic hydroxynonenal induced cell death of diverse brain neurons and hepatocytes of the Japanese macaque monkeys [18]. To elucidate the etiology of the Langerhans cell degeneration/death causing impaired insulin secretion, we made an appropriate monkey model for an increased blood concentration of hydroxynonenal to analyse its toxic effects upon the Langerhans islet. We focused on the pancreas of monkeys to elucidate the molecular cascade processing in the primate organ. The pancreas of monkeys is large enough to supply sufficient amount of tissues necessary for the simultaneous analyses of immunofluorescence histochemistry, light and electron microscopy, and Western blotting in the given animal. Here, we report that hydroxynonenal can induce lysosomal membrane rupture/permeabilization by activated calpain-mediated cleavage of the carbonylated Hsp70.1 for the occurrence of Langerhans cell degeneration/death.

## Materials and methods

### Animals

After the referee of animal experimentation about the ethical or animal welfare, four young (4~5 years: compatible with teenagers in humans) female Japanese macaque monkeys (*Macaca fuscata*) were supplied by National Bio-Resource Project (NBRP) "Japanese monkey" (National Institute for Physiological Sciences, Okazaki, Japan). After arrival, the monkeys were reared in the wide cage with autofeeding and autodrainage machines as well as appropriate toys to play at least for 1 year to facilitate acclimation. The room temperature was kept 22~24˚C with the humidity of 40~50%. They were fed by 350 kCal/Kg body weight of non-purified solid monkey foods per day containing vitamines. and apples, pumpkins or sweet potatoes were given twice every week. In the morning and afternoon, an animal care staff and the first author monitored the health and well-being of the animals to check the consumption of foods, pupilar reflex to the light, and conditions of standing and jumping.

At 5~6 years of age, monkeys with body weight 5~7 Kg were randomly divided into two different groups of the sham-operated control (n = 1) and those undergoing hyroxynonenal injections (n = 3). In 3 monkeys, under the intramuscular anesthesia using 2 mg/Kg of ketamine hydrochrolide, intravenous injections of 5 mg/week of synthetic hydroxynonenal (Cayman Chemical, Michigan, USA) were done for 24 weeks. Such doses and serial injections were designed to temporarily mimic blood concentrations of hydroxynonenal in humans around 60's [34]. Behavioural changes such as reduced exploration, standing, and jumping as well as decreases of appetite and body weight were carefully monitored to implement humane endpoints. However, all the monkeys were fine until the end of final 24th hydroxynonenal injections.

### Tissue collection

Six months after the initial injection and within a couple of weeks after the final injection, the monkeys were immobilized by the intramuscular injection of 10 mg/Kg BW ketamine hydrochloride followed by the intravenous injection of 50 mg/Kg BW sodium pentobarbital. In addition, to ameliorate animal suffering, the monkey was deeply anesthetized with 1.5% halothane plus 60% nitrous oxide. After the perfusion of 500 mL saline through the left ventricle, the pancreas was removed without suffering any pain. Half of the tissue was fixed in either i) 4%

paraformaldehyde for light microscopy, or ii) 2.5% glutaraldehyde for electron microscopy. The remaining half was stocked in the -80˚C deep freezer for the Western blotting analysis.

## Histological and immunofluorescence histochemical analyses

The pancreas tissues after fixation with 4% paraformaldehyde for 2 weeks were embedded in paraffin, and 5μm sections were stained by hematoxylin-eosin. For the immunofluorescence histochemistry, the cryoprotected pancreas tissues embedded in the OCT medium (Sakura Finetek, Japan) were cut by cryotome (Tissue-Tek® Polar®, Sakura, Japan), and 5μm sections were immersed with heated 0.01% Citrate retrieval buffer to induce epitope retrieval. Non-specific staining was blocked with 1% bovine serum albumin (Nacalai tesque, Japan), and were incubated overnight at 4˚C with the primary antibodies at the dilution of 1:100. We used mouse monoclonal anti-human Hsp70 (BD Bioscience, USA), rabbit anti-human activated μ-calpain (order made by PEPTIDE Institute, Japan), rabbit anti-human cathepsin B (Cell Signaling, USA), and mouse monoclonal anti-Lamp2 (Abcam, USA) antibodies. After washings, the sections were incubated for 30 min with secondary antibodies; Alexa Fluor™ 594 goat anti-mouse IgG [H+L] (Invitrogen, USA), or Alexa Fluor™ 488 goat anti-rabbit IgG (Invitrogen, USA) at the dilution of 1:500. To block autofluorescent staining, Autofluorescence Quenching Kit (Vector Laboratories, USA) was utilized. The immunoreactivity was observed with the laser confocal microscope (LSM5 PASCAL, Software ZEN 2009, Carls Zeiss, Germany).

## Ultrastructural analyses

For the electron microscopic analysis, small specimens of the pancreas tissue were fixed with 2.5% glutaraldehyde for 2h and 1% $OsO_4$ for 1h. Subsequently, they were dehydrated with graded acetone, embedded in resin (Quetol 812, Nisshin EM Co. Tokyo), and thin sections were made. After trimming with 0.5% toluidine blue-stained sections, the ultrathin (70nm) sections of appropriate portions were stained with uranyl acetate (15 min) and lead citrate (3 min), and were observed by the electron microscope (JEM-1400 Plus, JEOL Ltd., Tokyo).

## Western blotting

Total protein extraction was done, using protease inhibitor cocktail (Sigma-Aldrich, USA) and PhosSTOP phosphatase inhibitor cocktails tablets (Roche, Germany). After centrifugation at 12,000 rpm for 10 min, the supernatant proteins were determined by Bradford Assay (Thermo Fisher, USA). Twenty μg proteins were separated by SDS-PAGE in SuperSep (TM) Ace 5–20% gel (Wako, Japan) at 40 mA for 1h. The total proteins were transferred to PVDF membrane (Millipore, USA). Transferred protein quantities were detected with Ponseau S solution (Sigma-Aldrich, USA). Transferred proteins were blocked with 1% BSA (KPL Detector™ Block, USA) for 1h. The blots were incubated with mouse monoclonal anti-human Hsp70 antibody (BD Bioscience, USA) at the dilution of 1:4,000, rabbit anti-human activated μ-calpain antibody (PEPTIDE Institute, Japan) at 1:250 overnight, or rabbit anti-human GPR109A (also called hydroxycarboxylic acid receptor 2) antibody (Abcam, USA) at 1:500. β-actin was utilized as an internal control at a dilution of 10,000 (Sigma-Aldrich, USA). The immunoblots were subsequently incubated for 1h with secondary antibodies at 1:10,000 dilution of anti-mouse (Santa Cruz, USA) or anti-rabbit IgG (Sigma, USA). An enhanced chemiluminescence (ECL) HRP substrate detection kit (Millipore, USA) was used to visualize the reactive protein bands with ImageQuant LAS 4000 mini (GE Life Science, USA).

## Ethics

This study was carried out in strict accordance with the recommendations in the Guide for the Care and Use of Laboratory Animals of the National Institutes of Health. The protocol was approved by the Committee on the Ethics of Animal Experiments of the Kanazawa University Graduate School of Medical Sciences (Protocol Number: AP-153613).

## Results

By light microscopy, the Langerhans islets after the hydroxynonenal injections showed formation of many vacuoles, compared to the control, which were often filled with eosinophilic substance or looked almost empty (Fig 1). Nuclear chromatin of Langerhans cells was generally more dense than the control, which was consistent with the electron microscopic finding (Fig 2). A small number of nuclei showed dissolution of chromatin or punctuate condensation (Fig 1, dot circle). However, neither apoptotic bodies nor membrane blebbings were observed in the Langerhans islet cells after the hydroxynonenal injections.

The Langerhans islets are known to be basically composed of α-, β-, δ- cells and pancreatic polypeptide cells (PP-cells). As described previously in humans [35, 36], electron microscope was useful for differentiating 4 types of cells in the Langerhans islets of monkeys. In β-cells (β), insulin secretory granules had an electron-opaque core of 300–400 nm with a clear halo

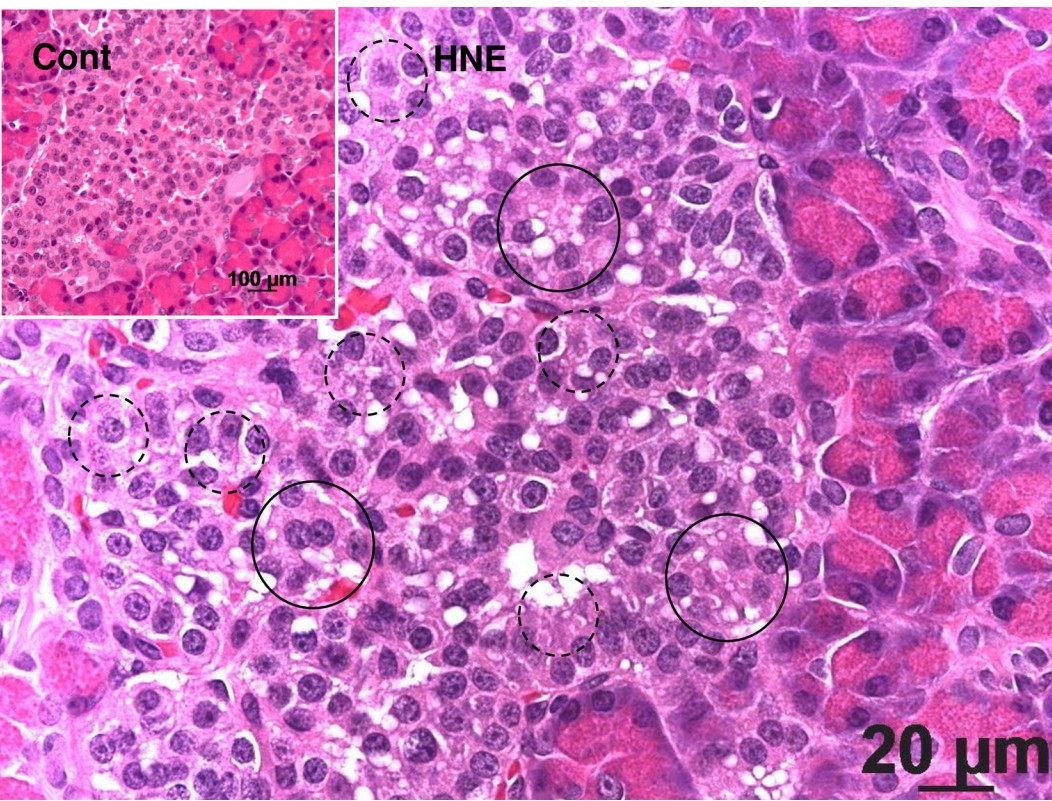

**Fig 1. Light micrograph of the monkey pancreas.** The Langerhans islet after the hydroxynonenal injections (HNE) shows many vacuole formation (circles), compared to the sham-operated control (Cont). Nuclear chromatin is generally more dense after the hydroxynonenal injections, compared to the control. A small number of nucleus shows diffuse dissolution or punctuate condensation (dot circle). However, neither apoptotic bodies nor membrane blebbings were seen. Acinar cells are distributed in the surrounding area. Hematoxylin-eosin staining.

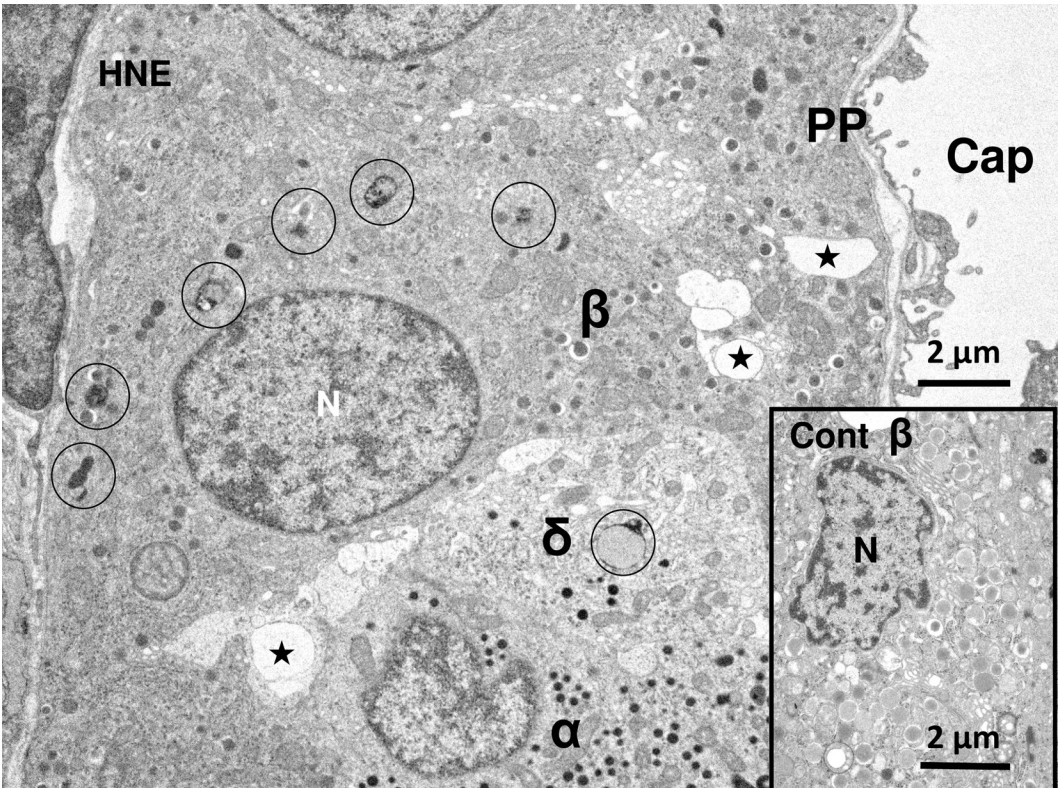

**Fig 2. Electron micrograph of the Langerhans islet of the monkey before and after the hydroxynonenal injections.** The Langerhans islet cells are situated surrounding a capillary (Cap), and each cell is characterized by peculiar secretory granules. Insulin secretory granules of the β-cell (β) are the largest with a clear halo. Glucagon secretory granules of the α-cell (α) lack a clear halo, being a little bit smaller but more electron-dense than insulin granules. Somatostatin-containing granules of the δ-cell (δ) are also electron-dense like glucagon granules, showing a sparse distribution. PP-cells (PP) contain spherical or elliptical small granules, which are very heterogeneous in size. The greatest changes after the hydroxynonenal injections are a remarkable decrease of insulin and somatostatin granules as well as an increase of autophagosomes and autolysosomes. In the control β-cell (Cont), insulin granules were distributed throughout the cytoplasm. In contrast, in the β-cell after the hydroxynonenal injections (HNE), the distribution of insulin granules was restricted in the cytoplasm toward the capillary. Instead, especially β-cell showed formation of many autolysosomes (circles), compared to the control (Cont). Further, both β-cell and δ-cell showed vacuole formations (stars) and more electron-dense nuclear chromatin. N; nucleus.

(Figs 2–5), while in α-cells (α), glucagon secretory granules were electron-dense without a clear halo. Compared to the insulin secretory granules, the glucagon granules were more electron-dense with a smaller diameter (200–300 nm) (Fig 2). δ-cells (δ) exhibit neuron- or trumpet- like morphology with cytoplasmic processes extending from the islet capillaries (Fig 2). Somatostatin-containing granules were also electron-dense, showing a similar size with glucagon granules, but showed a sparse distribution (Fig 6A and 6B). PP-cells (PP) contained spherical smaller granules, which were heterogeneous in size (Fig 2).

By the electron microscopic observation, the most remarkable change in the Langerhans cells after the hydroxynonenal injections was a remarkable decrease of insulin and somatostatin granules (Fig 2, HNE), compared to the control (Fig 2, Cont). The decrease was evident, when compared with human β- and δ- cells whose cytoplasm was filled with insulin and somatostatin granules [36]. In the control β-cells of monkeys also, insulin granules were distributed throughout the cytoplasm (Fig 2, Cont), whereas in the Langerhans islets after the hydroxynonenal injections, they were clustered in the cytoplasm toward the capillary lumen

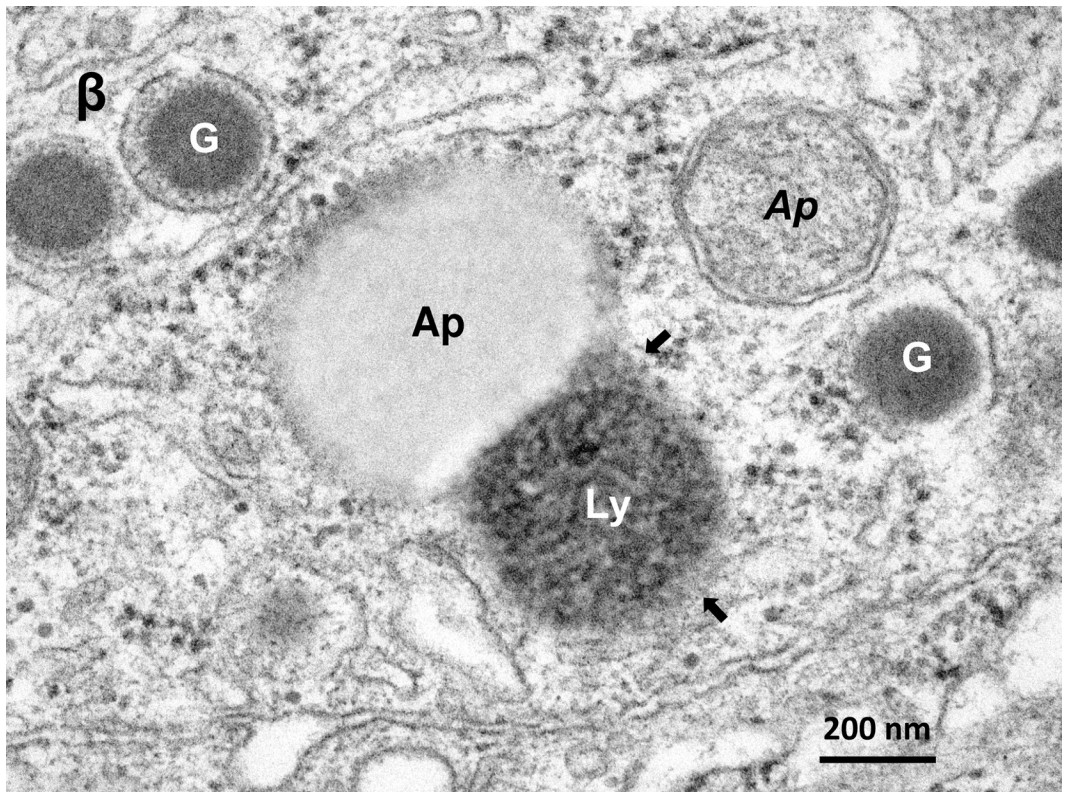

**Fig 3. Electron micrograph of the β-cell in the monkey after the hydroxynonenal injections. β-cell** shows fusion of an autophagosome (AP) with a lysosome (Ly) exhibiting evidence of membrane permeabilization (arrows). This autophagosome is devoid of double membrane, whereas another one (*Ap*) presumably containing degenerated mitochondria, has distinct double membrane. G: insulin granule.

(Fig 2, HNE). Among α-, β-, δ- and PP- cells of the islets after the hydroxynonenal injections, β- and δ- cells showed lots of vacuole formations which were revealed to be enlarged rough ER with a marked decrease of ribosomes (Fig 2 stars, Fig 5), indicating ER dysfunction. Further, β- and δ- cells after the hydroxynonenal injections were ultrastructurally characterized by peroxisomal proliferation (Figs 5 and 6B), and increments of autophagosomes or autolysosomes (Figs 2–5, 6A and 6B), which were extremely rare in the control tissue (Fig 2, Cont).

By the electron microscopic observation of the pancreas, lysosomes were relatively electron-dense organelles homogenously filled with microvesicular granules, while peroxisomes contained a less-dense matrix with fine granules. So, the limiting membrane was more clearly visible in peroxisomes than lysosomes. In the β-cells after the hydroxynonenal injections, lysosomes were often distributed around the autophagosomes or autolysosomes throughout the cytoplasm (Figs 2 and 3), while peroxisomes were distributed around insulin granules, being intermingled with degenerating mitochondria in the cytoplasm toward the capillary lumen (Fig 5). At the high magnification, degenerating β-cells (Fig 3) and δ-cells (Fig 6A) contained an autophagosome which was in the process of fusing with the lysosome showing membrane permeabilization. Autophagosome or autolysosome was observed to fuse also with peroxisome which was surrounded by the disrupted membrane (Figs 4 and 6B). Peroxisomes with membrane disruption were abundant around insulin and somatostatin granules in the cytoplasm toward the capillary lumen (Figs 4 and 6A). Some peroxisomes were seen to fuse with degenerating

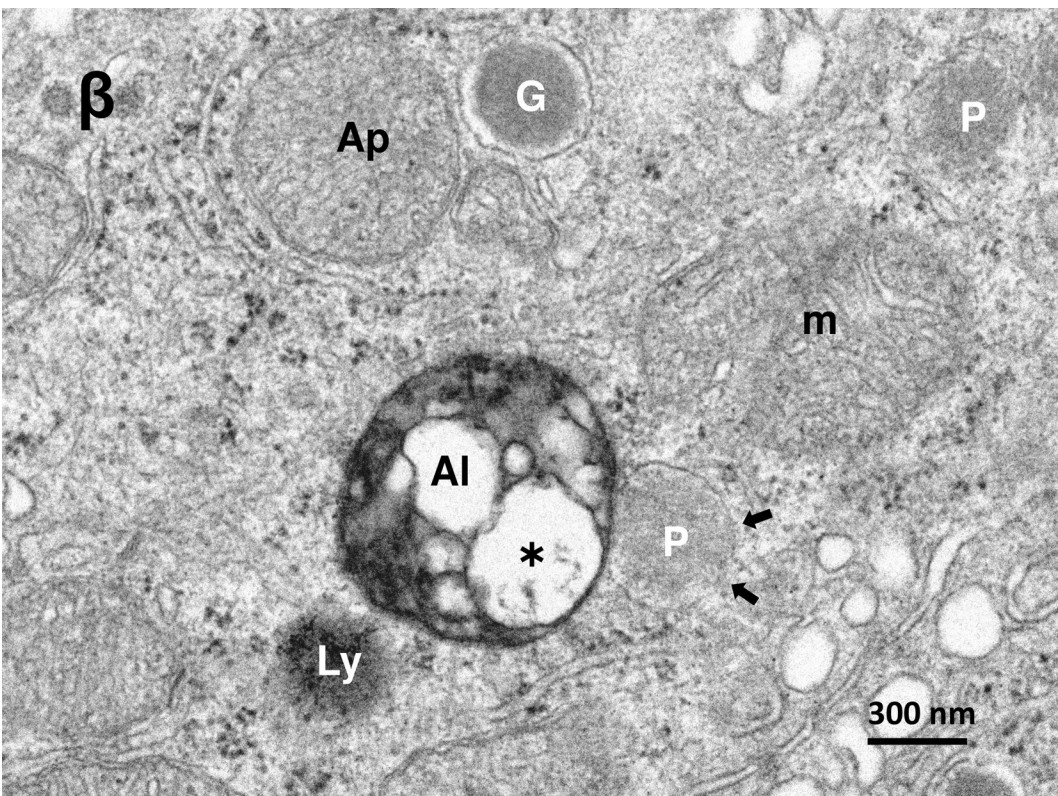

**Fig 4. Electron micrograph of the β-cell in the monkey after the hydroxynonenal injections.** β-cell shows fusion of an autolysosome (Al) with a peroxisome (P) exhibiting membrane disruption (arrows). The autolysosome (Al) contains mitochondria-derived debris (asterisk). The neighbouring lysosome (Ly) shows membrane permeabilization. G: insulin granule, Ap: autophagosome, m: degenerating mitochondria.

mitochondria (Fig 5, circles) or in the close vicinity of autolysosomes (Fig 5, dot circles). As β-cells and δ-cells showed a similar change, β-cells generally showed a stronger degeneration than δ-cells.

Immunofluorescence histochemical analysis showed a remarkable increase of Hsp70.1 colocalization with activated μ-calpain in the Langerhans islets after the hydroxynonenal injections (Fig 7A, HNE, merged color), comparted to the control (Fig 7A, Cont). In the control, the cathepsin B immunoreactivity was seen as tiny granules which were compatible with the size of intact lysosomes (Fig 7B, Cont). However, after the hydroxynonenal injections, cathepsin B immunoreactivity was seen as coarse granules with a diffuse, thin distribution throughout the cytoplasm (Fig 7B, HNE). In the core of cathepsin B immunoreactivity, lamp-2 immunoreactivity was colocalized with a merged color (data not shown here), which showed extralysosomal release of cathepsin enzymes by the lysosomal membrane rupture/permeabilization.

Western blotting showed an increase of 76 kDa band intensities of activated μ-calpain after the hydroxynonenal injections, compared to the control (Fig 7C, rectangle). In response to such stress, the main bands of Hsp70.1 are increased remarkably, compared to the control (Fig 7D, rectangle). In addition, calpain-mediated Hsp70.1 cleavage [33], as shown in 30 kDa bands (Fig 7D, dot rectangle), are increased in the monkeys after hydroxynonenal injections. Repeated Western blotting showed the same results, although the statistical significance was not available because of the restricted number of experimental animals. GPR109A was expressed equally in the control and hydroxynonenal-injected tissues (Fig 7C).

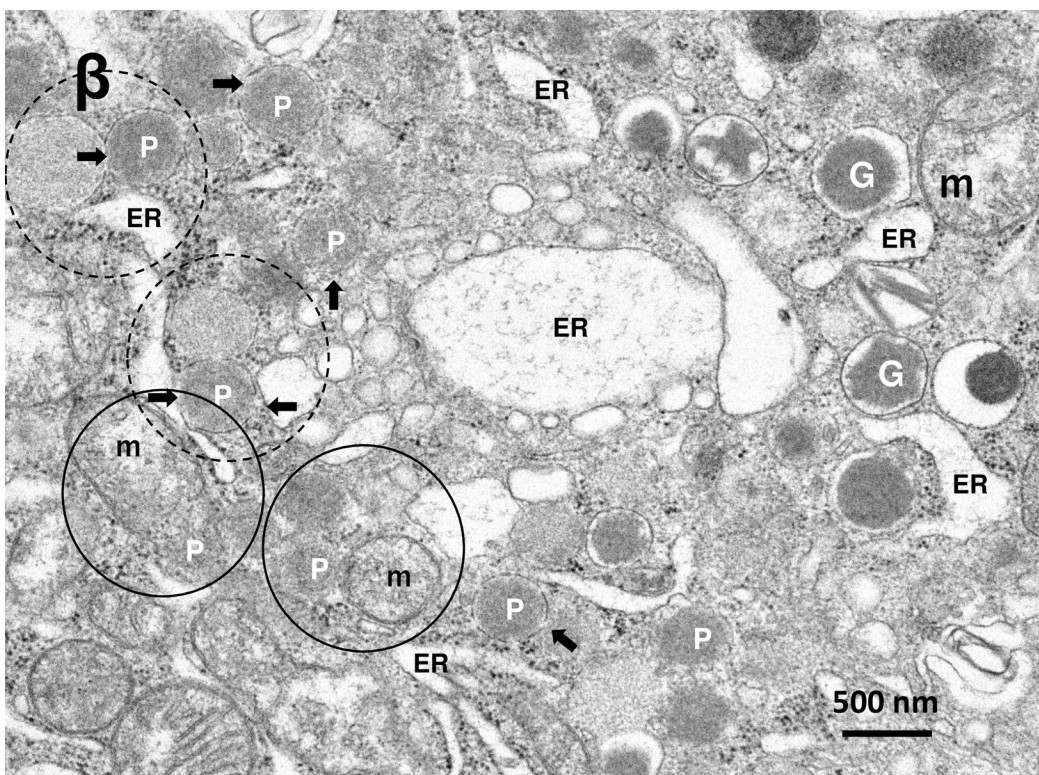

**Fig 5. Electron micrograph of the β-cell in the monkey after the hydroxynonenal injections.** β-cell shows enlargement of rough ER (ER), as seen in Fig 2. Proliferation of peroxisomes (P) is obvious, and they are intermingled with degenerating mitochondria and insulin granules in the cytoplasm toward the capillary lumen, as seen in Fig 2. Some peroxisomes with membrane disruptions (arrows) are fusing with degenerating mitochondria (m, circles) or autolysosomes presumably containing mitochondrial debris (dot circles). G: insulin granule.

## Discussion

In type 2 diabetes, β-cell degeneration and death contribute to the loss of its functional mass [8, 37, 38]. However, surprisingly, the underlying molecular mechanism remains grossly unknown until now, although β-cell lipotoxicity has been subject to intensive research for the past two decades [39–41]. As most studies showing deleterious effects of elevated NFFAs on β-cell function were conducted *in vitro*, the contribution of elevated NEFAs to β-cell dysfunction in human patients with type 2 diabetes still remains controversial [41]. Not only quantity but also quality (i.e. extent and type of oxidization) of NEFAs may impact β-cell function [10].

In the pancreas, two types of fatty acid receptor emerge as a cause of abnormal $Ca^{2+}$ mobilization in response to excessive and/or oxidized NEFAs; one is G protein-coupled receptor 40 (GPR40) while another is GPR109A. GPR40 is a receptor for diverse NEFAs, being expressed abundantly in the β-cells [42, 43]. Steneberg et al. (2005) [44] proposed that GPR40 is indirectly responsible for hyperinsulinemia-induced insulin resistance, whereas other researchers reported that NEFA-induced hyperinsulinemia represents a mechanism by which β-cells attempt to compensate for the insulin resistance because this ability is compromised by GPR40 deletion [45, 46]. Accordingly, implication of GPR40 for the development of β-cell lipotoxicity in response to excessive NEFAs still remains debated [5]. Albeit GPR40 is abundantly expressed in the primate pancreas [42, 43] including the present macaque monkeys (data not shown), for this reason we focused here on the expression of a metabolic sensor GPR109A in the monkey pancreas. Since

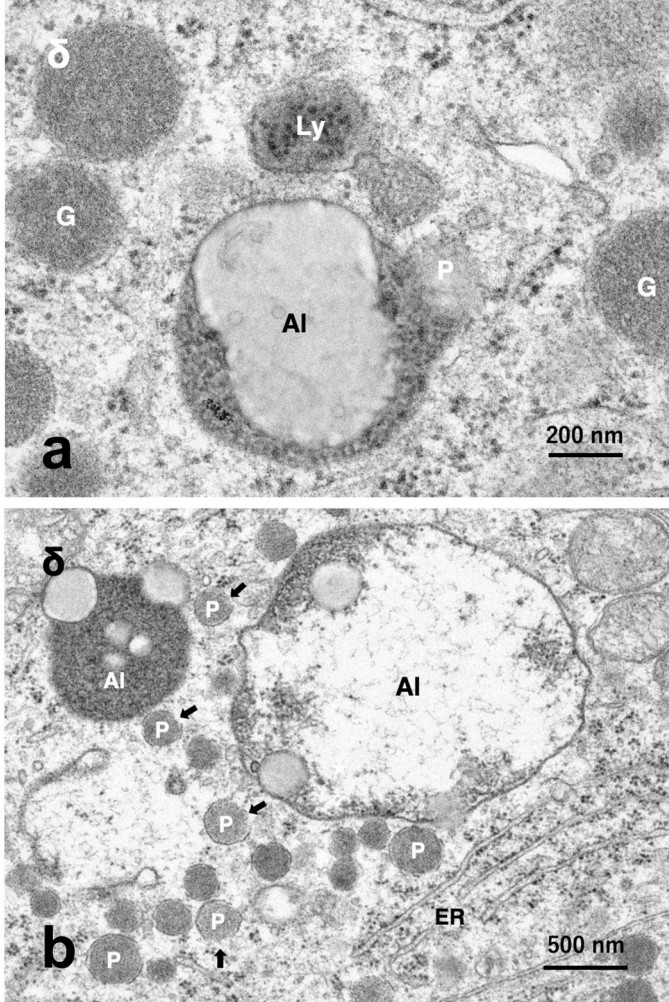

**Fig 6.** a) Electron micrograph of the δ-cell in the monkey after the hydroxynonenal injections. **δ-cell** shows fusion of an autolysosome (Al) with a peroxisome (P) exhibiting membrane disruption. G: somatostatin granule without a halo, Ly: lysosome. b) Electron micrograph of the δ-cell in the monkey after the hydroxynonenal injections. δ-cell shows peroxisomal proliferation (P) around the autolysosomes (Al). The latter are presumably in the process of fusing with peroxisomes (P). Some peroxisomes show membrane disruption (arrows). ER: rough ER.

in response to hydroxynonenal GPR109A was demonstrated to induce excessive $Ca^{2+}$ mobilization and the subsequent cell death in the retinal pigmented and colon epithelial cells [47], we speculated that the same may occur in β-cells where GPR109A is expressed.

Previous reports suggest that oxidative stress during metabolizing excessive NEFAs mediates lipotoxicity. As NEFAs are metabolized not only through mitochondrial β-oxidation but also through peroxisomal β-oxidation, the subcellular sites of ROS formation are mitochondria and peroxisomes [27, 28, 48]. In contrast to the mitochondrial β-oxidation, however, the acyl-CoA oxidases generate more $H_2O_2$ in the peroxisomes [49]. Now, almost 50 years after their discovery, it is well known that peroxisomes can function as a main source of ROS [50]. As β-cells almost completely lack catalase, they are thought to be exceptionally vulnerable to abundant $H_2O_2$ which was presumably generated in the increased number of peroxisomes (Fig 5).

It is well known that peroxisomes are highly plastic, dynamic organelles that rapidly modulate their size, number, and enzyme content in response to changing environmental conditions

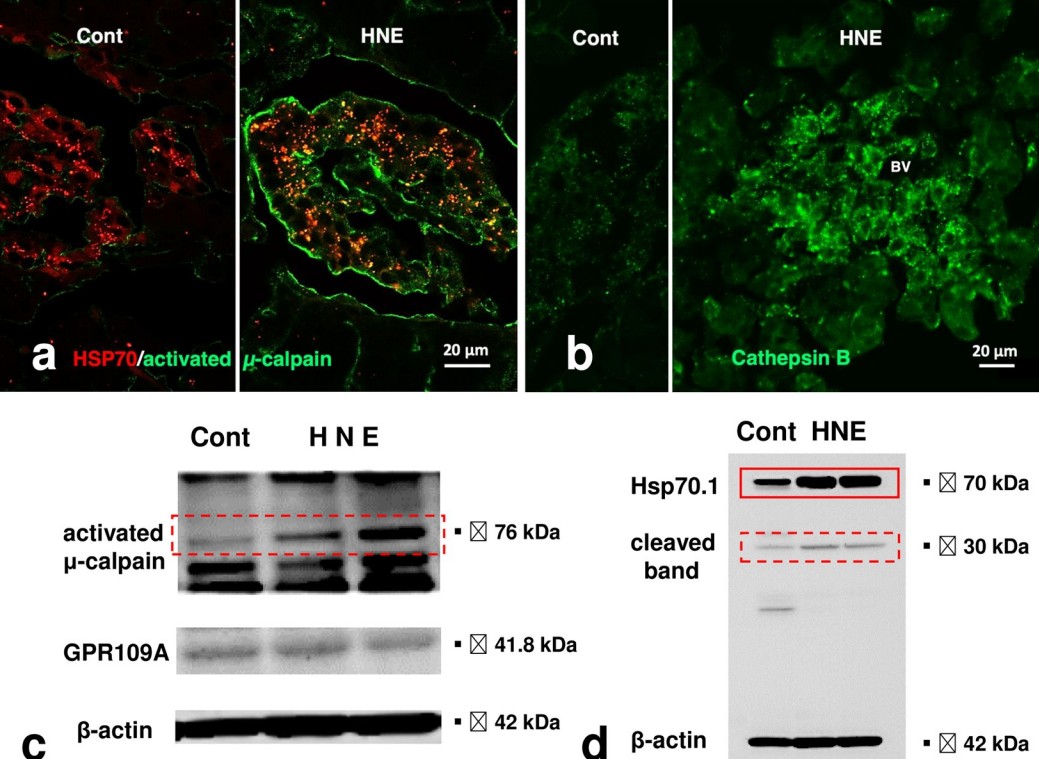

**Fig 7. Calpain activation, Hsp70.1 cleavage, and cathepsin B leakage.** a) Immunofluorescence histochemical staining of Hsp70.1 (red) and activated μ-calpain (green). Activated μ-calpain immunoreactivity is negligible without hydroxynonenal injections (Cont), whereas μ-calpain activation occurs with hydroxynonenal injections (HNE), being consistent with the Western blotting data (c, activated μ-calpain). After hydroxynonenal injections, activated μ-calpain immunoreactivity (green) is colocalized with Hsp70.1 immunoreactivity (red), showing a merged color of yellow (HNE, yellow). The distribution pattern of granular merged colors is compatible with that of autolysosomes as seen in Fig 2. b) Immunofluorescence histochemical staining of cathepsin B (green). Cathepsin B is stained as tiny granules in the control Langerhans islet (Cont), whereas stained as coarse granules with the perigranular immunoreactivity after hydroxynonenal injections (HNE), which indicates lysosomal membrane rupture/permeabilization. The nuclei show negligible cathepsin B immunoreactivity. Cathepsin B immunoreactivity was colocalized with that of Lamp2 (data not shown here). c, d) Western blotting analyses of activated μ-calpain (c, 76 kDa), GPR109A (c, 41.8 kDa), and Hsp70.1 (d, 70 kDa). Compared to the control (c, Cont), μ-calpain is activated after hydroxynonenal injections (c, HNE). In response to the cell stress due to hydroxynonenal injections (d, HNE), not only Hsp70.1 main bands (d, rectangle) but also cleaved Hsp70.1 bands of 30 kDa (d, dot rectangle) are increased, compared to the control (d, Cont). GPR109A is expressed in the pancreas tissue, although showing no change after hydroxynonenal injections (c, GPR109A).

[51, 52]. The present study showed a remarkable peroxisomal proliferation in β-cells after the hydroxynonenal injections (Figs 5 and 6B). Inactivation of $H_2O_2$ by catalase seems to be a step of critical importance for the removal of ROS in β-cells. However, since β-cells almost completely lack the $H_2O_2$-detoxifying enzyme, oxidoreductase catalase [1, 2], they are exceptionally vulnerable to $H_2O_2$ that was generated in peroxisomes. If $H_2O_2$ is not quickly converted into water and oxygen by catalase, it can react in an iron-catalysed reaction with $O_2\bullet$ – yielding the highly reactive $OH\bullet$. It is the most reactive oxygen radical known, reacting instantaneously with molecules in its immediate vicinity, which explains its great destructive power. This conceivably causes β-cell dysfunction, degeneration, and ultimately cell death, because of its low antioxidative defense status [23]. Since mitochondria also lack catalase and, in β-cells, mitochondria contain an extremely low glutathione peroxidase activity, any excessive oxidative stress is detrimental for the β-cell mitochondria due to the limited antioxidative defence capacity [19]. The capacity for inactivation of $H_2O_2$ is sufficient under normal

circumstances, however, during the consecutive oxidative stress like the present experimental paradigm (5mg/W x 24 week injections of hydroxynonenal), a constant level of $H_2O_2$ might have been continuously generated within both mitochondria and peroxisomes. So, low anti-oxidative defence of β-cells can be easily overwhelmed in pathological situations of sustained OH• production. The close spatial relation of mitochondria with crista disruption and peroxisome with membrane disruption (Fig 5), indicates a strong impact of hydroxynonenal upon β-cells.

Peroxisomal degradation occurs through at least three different mechanisms: macropexophagy, micropexophagy, and 15-lipoxygenase-mediated autolysis [53, 54]. A lipid-peroxidizing enzyme, 15-lipoxygenase can generate hydroxynonenal from ω-6 polyunsaturated fatty acids of biomembranes. As 15-lipoxygenase is distributed in peroxisomes at the highest concentration [55], it initiates organelle degradation by pore formation in the limiting membranes [56, 57]. van Leyen et al. (1998) [58] demonstrated that 15-lipoxygenase can bind to organelle membranes and induce the leakage of its contents. Yokota et al. (2001) [55] also showed induction of the peroxisomal membrane disruption and the resultant catalase leakage by 15-lipoxygenase in the rat liver. Interestingly, such 15-lipoxygenase-induced peroxisomal membrane disruptions were very similar at the ultrastructural level to those observed in the monkey Langerhans cells after the hydroxynonenal injections (Figs 5 and 6B). Hydroxynonenal is generated as a consequence of $H_2O_2$-mediated lipid peroxidation of biomembranes composing of ω-6 polyunsaturated fatty acids such as linoleic and arachidonic acid [11, 59, 60]. Therefore, it is reasonable to speculate that exogenous hydroxynonenal is a causative substance and also a trigger, which induce consecutive peroxisomal HO• production with the resultant intrinsic hydroxynonenal generation in the monkey pancreas (Fig 8). Hydroxynonenal has been increasingly recognized as a particularly important mediator of dysfunction and degeneration of pancreatic β cells in type 2 diabetes and alcohol-induced pancreatic damage [19, 61]. There is ample evidence that hydroxynonenal causes cell dysfunction and death in diverse disorders including Alzheimer's disease [62, 63], cardiovascular disease, stroke, arthritis and asthma [64–67]. By activating GPR109A, hydroxynonenal can be a trigger for the excessive $Ca^{2+}$ mobilization and subsequent calpain activation [68]. So, it is reasonable that the Langerhans cells with GPR109A expression showed an increased activation of μ-calpain in response to the exogenous hydroxynonenal.

Concerning the molecular mechanism of ischemic neuronal death, Yamashima et al. formulated the 'calpain-cathepsin hypothesis' in 1998 [14]. Thereafter, concerning the mechanism of Alzheimer's neuronal death, they suggested that Hsp70.1 with dual functions of

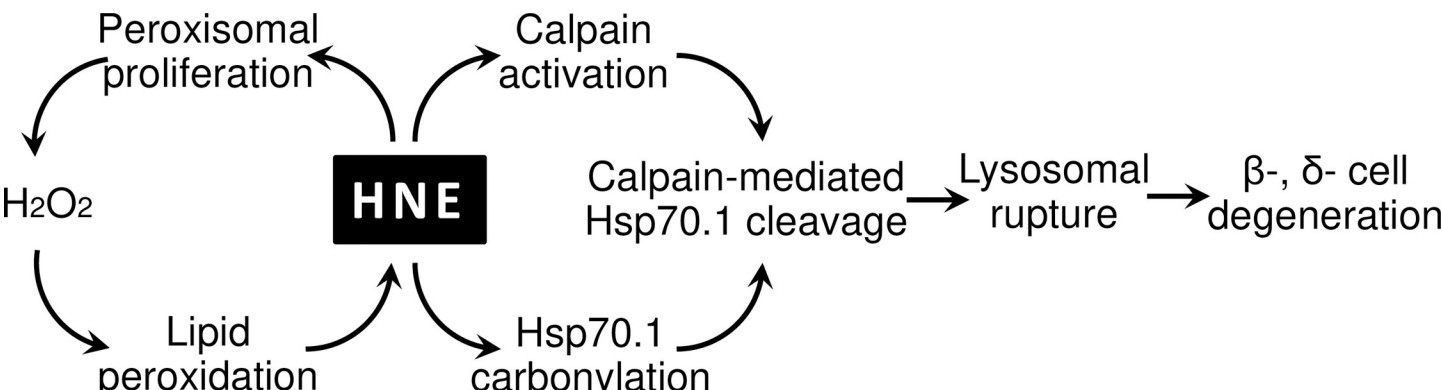

**Fig 8. The molecular cascade explaining degeneration of β- and δ- cells in the Langerhans islet of the monkey after hydroxynonenal injections.** Peroxisomes play a crucial role for the sustained $H_2O_2$ production, while calpain-mediated cleavage of carbonylated Hsp70.1 causes lysosomal membrane rupture/permeabilization. In both cascades, hydroxynonenal plays a central role for the β- and δ- cell degeneration.

molecular chaperone and lysosomal stabilizer becomes vulnerable to the cleavage by activated μ-calpain, which is facilitated after Hsp70.1 oxidation (carbonylation) by hydroxynonenal [33, 69–71]. Elevated hydroxynonenal was recently reported to be responsible for hyperglycemia and correlated with HbA1c in both experimental animals and humans [72], although the precise role of hydroxynonenal in type 2 diabetes is not well understood until now. Due to the hydroxynonenal-induced carbonylation with the subsequent calpain-mediated cleavage of carbonylated Hsp70.1, functional Hsp70.1 decreases steadily. This results in both accumulation of autophagosomes and permeabilization/rupture of the lysosomal membrane (Fig 8). It is likely that autophagy deficiency and lysosomal failure are occurring simultaneously. By inducing Hsp70.1 disorder, hydroxynonenal conceivably plays sinister roles in the occurrence of degeneration/death of Langerhans cells, leading to type 2 diabetes.

Applying the data of the present monkey experimental paradigm to consider human diseases, both the dietary ω-6 fatty acid- (exogenous; from cooking oils, deep-fried foods, etc.) and the biomembrane phospholipid- (intrinsic; from the circumstance containing electromagnetic wave, air pollution, etc.) peroxidation product hydroxynonenal, when combined, conceivably plays crucial roles for the lysosomal cell death not only in the pancreas but also in the brain. The authors speculate that not only type 2 diabetes but also Alzheimer's disease might occur in response to the exogenous and intrinsic hydroxynonenal by the long-standing lysosomal cell death in the corresponding organ [11].

## Conclusions

During the last four decades, hydroxynonenal, a major α, β-unsaturated aldehyde product of ω-6 fatty acid oxidation, has been shown to be involved in a great number of pathologies such as metabolic diseases, neurodegenerative diseases, and cancers. The present study indicated that a major pathophysiological mechanism behind the development of Langerhans cell degeneration/death is oxidative stress mediated by hydroxynonenal. By carbonylating a lysosomal stabilizer, Hsp70.1, hydroxynonenal contributes to the occurrence of lysosomal membrane rupture/permeabilization in the Langerhans cells, which lead to their degeneration/death. Calpain activation by hydroxynonenal may contribute to cleavage of the oxidized (carbonylated) Hsp70.1. In case of the above diseases, chronic ischemia and/or inflammation may also facilitate calpain activation. The low defense against exogenous hydroxynonenal and peroxisomally-generated hydrogen peroxide, was presumably overwhelmed in the Langerhans islets. The authors speculate from the present monkey data that 'hydroxynonenal' might be a real culprit behind type 2 diabetes in humans. Future studies focusing on the earlier phase of hydroxynonenal toxification in vivo, are needed to elucidate the molecular mechanism of the Langerhans cell pathology for the development of type 2 diabetes.

## Supporting information

**S1 Data.**
(PDF)

## Acknowledgments

The authors are deeply indebted to Mr. Jun Uchimoto for the daily care of monkeys and autopsy assistance, Mrs. Katsumi Hara and Mrs. Masayo Baba for the tissue preparation, and Mrs. Rie Nishioka and Mrs. Mai Nakayama for the secretory work.

## Author Contributions

**Conceptualization:** Tetsumori Yamashima.

**Funding acquisition:** Tetsumori Yamashima.

**Methodology:** Piyakarn Boontem.

**Writing – original draft:** Piyakarn Boontem.

**Writing – review & editing:** Tetsumori Yamashima.

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
