## [Decision Letter · Decision Letter 0]

8 Sep 2021

PONE-D-20-37303

Hydroxynonenal causes Langerhans cell degeneration in the pancreas of Japanese macaque monkeys

PLOS ONE

Dear Dr. Yamashima,

Thank you for submitting your manuscript to PLOS ONE.  Thank you for your patient whilst there has been a change in handling editor, and a preference from the editorial office to source reviewers with experience on using  monkeys as the model.

After careful consideration, we feel that it has merit but does not fully meet PLOS ONE’s publication criteria as it currently stands. Therefore, we invite you to submit a revised version of the manuscript that addresses the points raised during the review process.

We look forward to receiving your revised manuscript.

Kind regards,

David Chau

Academic Editor

PLOS ONE

Journal Requirements:

3. PLOS requires an ORCID iD for the corresponding author in Editorial Manager on papers submitted after December 6th, 2016. Please ensure that you have an ORCID iD and that it is validated in Editorial Manager. To do this, go to ‘Update my Information’ (in the upper left-hand corner of the main menu), and click on the Fetch/Validate link next to the ORCID field. This will take you to the ORCID site and allow you to create a new iD or authenticate a pre-existing iD in Editorial Manager. Please see the following video for instructions on linking an ORCID iD to your Editorial Manager account: https://www.youtube.com/watch?v=_xcclfuvtxQ"

5. PLOS ONE now requires that authors provide the original uncropped and unadjusted images underlying all blot or gel results reported in a submission’s figures or Supporting Information files. This policy and the journal’s other requirements for blot/gel reporting and figure preparation are described in detail at https://journals.plos.org/plosone/s/figures#loc-blot-and-gel-reporting-requirements and https://journals.plos.org/plosone/s/figures#loc-preparing-figures-from-image-files. When you submit your revised manuscript, please ensure that your figures adhere fully to these guidelines and provide the original underlying images for all blot or gel data reported in your submission. See the following link for instructions on providing the original image data: https://journals.plos.org/plosone/s/figures#loc-original-images-for-blots-and-gels. In your cover letter, please note whether your blot/gel image data are in Supporting Information or posted at a public data repository, provide the repository URL if relevant, and provide specific details as to which raw blot/gel images, if any, are not available. Email us at plosone@plos.org if you have any questions

Reviewers' comments:

Reviewer's Responses to Questions

**Comments to the Author**

1. Is the manuscript technically sound, and do the data support the conclusions?

Reviewer #1: Yes

Reviewer #2: Yes

2. Has the statistical analysis been performed appropriately and rigorously? 

Reviewer #1: I Don't Know

Reviewer #2: Yes

3. Have the authors made all data underlying the findings in their manuscript fully available?

Reviewer #1: Yes

Reviewer #2: Yes

4. Is the manuscript presented in an intelligible fashion and written in standard English?

Reviewer #1: Yes

Reviewer #2: Yes

5. Review Comments to the Author

Reviewer #1: The rational behind the experiment was clear and straight forward. The manuscript is almost well written.

While many different sources are used to set up the study in the introduction, little previous evidence is

stated. The introduction is thus short and poorly sets up the rationale for the study. More attention to how

this study fits into previous work in oxidative and inflammation should be added to improve this section.

Please refer to doi: 10.3390/antiox9090781, 10.3390/antiox9100992

Reviewer #2: The authors investigated about the role of hydroxynonenal in both the carbonylation of Hsp70.1 and the activation of µ-calpain using Japanese macaque monkeys.

The rational behind the study was clear and straight forward. The manuscript is clearly written, its original and of interest in its field. However, the authors should better emphasize the conclusions.

I recommend that the paper be accepted with minor revision.

6. PLOS authors have the option to publish the peer review history of their article (what does this mean?). If published, this will include your full peer review and any attached files.

Reviewer #1: No

Reviewer #2: No

---

## [Author Response · Author response to Decision Letter 0]

15 Sep 2021

PONE-D-20-37303

Hydroxynonenal causes Langerhans cell degeneration in the pancreas of Japanese macaque monkeys 

PlosONE 

Editorial Board 

Dear Sirs, 

Sept 11, 2021 

This is to acknowledge for sending us two reviewer’s comments about the above paper. Their comments were extremely helpful for revising the manuscript.

As suggested by two reviewers, we have revised as followings. 

Reviewer 1: The rational behind the experiment was clear and straight forward. The manuscript is almost well written. While many different sources are used to set up the study in the introduction, little previous evidence is stated. The introduction is thus short and poorly sets up the rationale for the study. More attention to how this study fits into previous work in oxidative and inflammation should be added to improve this section.

# The highlights portion shows sentences added or revised in the INTRODUCTION. We have revised so as to strengthen the rational studying implication of hydroxynonenal for the development of Langerhans cell degeneration in type 2 diabetes.

The pathogenesis of β-cell dysfunction leading to insulin resistance and impaired insulin secretion seen in type 2 diabetes is incompletely understood. Previous animal and in vitro studies demonstrated that chronic levels of oxidative stress are among the earliest abnormalities in the natural history of type 2 diabetes, although the evidence is insufficient [6]. On the other hand, the evidence for inflammation in the etiology of type 2 diabetes is broad, however, the underlying mechanism is also not well understood. Increased oxidative stress and inflammation, combined together, may lead to the pancreatic β-cell failure [1,2,3,4,5]. Elevated levels of inflammatory markers, pro-oxidants and various markers of oxidative tissue damage were found in the diabetic patients, which indicates their involvement in the occurrence of insulin resistance and impaired insulin secretion. Peroxidation of polyunsaturated fatty acids is intensified in cells subjected to oxidative stress and inflammation, and it results in the generation of various bioactive compounds [6,7,8,9,10]. Reactive aldehyde, especially hydroxynonenal, is considered as the representative cytotoxic product of lipid peroxidation causing long-lasting biological consequences, in particular, by covalent modification of macromolecules. Hydroxynonenal is a major aldehyde which was produced during the OH•-mediated peroxidation of ω-6 polyunsaturated fatty acids. In 2009, Oikawa et al. (2009) [16] found by the proteomic analysis that Hsp70.1 can be a target of oxidation (carbonylation) by hydroxynonenal. Subsequently, Sahara and Yamashima (2010) [33] demonstrated by the in-vitro experiment that Hsp70.1 being carbonylated by hydroxynonenal is prone to activated μ-calpain-mediated cleavage. Calpain-mediated cleavage of carbonylated Hsp70.1 leads to the lysosomal membrane rupture/permeabilization with the resultant release of cathepsins and cell death. 

It is well-known that insulin resistance and impaired insulin signalling may be a contributory factor to the progression of type 2 diabetes. Although reduced release of insulin from β-cells is fundamental for both, the molecular mechanism of the Langerhans cell dysfunction caused by chronic inflammation and oxidative stress remains grossly unknown. Much progress has been made in identifying the molecular basis of lipotoxicity in the experimental systems, however whether or not this phenomenon actually plays an important role for the Langerhans cell degeneration remains debated especially in the primates [10]. To overcome this controversy, the authors thought that the experimental paradigm using non-human primates is indispensable. In the preliminary study, we confirmed that the intravenous injections of the synthetic hydroxynonenal induced cell death of diverse brain neurons and hepatocytes of the Japanese macaque monkeys [18]. To elucidate the etiology of the Langerhans cell degeneration/death causing impaired insulin secretion, we made an appropriate monkey model for an increased blood concentration of hydroxynonenal to analyse its toxic effects upon the Langerhans islet. We focused on the pancreas of monkeys to elucidate the molecular cascade processing in the primate organ. The pancreas of monkeys is large enough to supply sufficient amount of tissues necessary for the simultaneous analyses of immunofluorescence histochemistry, light and electron microscopy, and Western blotting in the given animal. Here, we report that hydroxynonenal can induce lysosomal membrane rupture/permeabilization by activated calpain-mediated cleavage of the carbonylated Hsp70.1 for the occurrence of Langerhans cell degeneration/death.

In addition, Ref 6 was changed, and Ref 72 was added.

Reviewer 2: The authors investigated about the role of hydroxynonenal in both the carbonylation of Hsp70.1 and the activation of µ-calpain using Japanese macaque monkeys. The rational behind the study was clear and straight forward. The manuscript is clearly written, its original and of interest in its field. However, the authors should better emphasize the conclusions.

# The CONCLUSION was completely revised to emphasize implication of ‘hydroxynonenal’ for the development of type 2 diabetes.

Conclusions

During the last four decades, hydroxynonenal, a major α,β-unsaturated aldehyde product of ω-6 fatty acid oxidation, has been shown to be involved in a great number of pathologies such as metabolic diseases, neurodegenerative diseases, and cancers. The present study indicated that a major pathophysiological mechanism behind the development of Langerhans cell degeneration/death is oxidative stress mediated by hydroxynonenal. By carbonylating a lysosomal stabilizer, Hsp70.1, hydroxynonenal contributes to the occurrence of lysosomal membrane rupture/permeabilization in the Langerhans cells, which lead to their degeneration/death. Calpain activation by hydroxynonenal may contribute to cleavage of the oxidized (carbonylated) Hsp70.1. In case of the above diseases, chronic ischemia and/or inflammation may also facilitate calpain activation. The low defense against exogenous hydroxynonenal and peroxisomally-generated hydrogen peroxide, was presumably overwhelmed in the Langerhans islets. The authors speculate from the present monkey data that ‘hydroxynonenal’ might be a real culprit behind type 2 diabetes in humans. Future studies focusing on the earlier phase of hydroxynonenal toxification in vivo, are needed to elucidate the molecular mechanism of the Langerhans cell pathology for the development of type 2 diabetes.

# The end of DISCUSSION was also modified a little bit. 

Concerning the molecular mechanism of ischemic neuronal death, Yamashima et al. formulated the ‘calpain-cathepsin hypothesis’ in 1998 [14]. Thereafter, concerning the mechanism of Alzheimer’s neuronal death, they suggested that Hsp70.1 with dual functions of molecular chaperone and lysosomal stabilizer becomes vulnerable to the cleavage by activated μ-calpain, which is facilitated after Hsp70.1 oxidation (carbonylation) by hydroxynonenal [33,69,70,71]. Elevated hydroxynonenal was recently reported to be responsible for hyperglycemia and correlated with HbA1c in both experimental animals and humans [72], although the precise role of hydroxynonenal in type 2 diabetes is not well understood until now. Due to the hydroxynonenal-induced carbonylation with the subsequent calpain-mediated cleavage of carbonylated Hsp70.1, functional Hsp70.1 decreases steadily. This results in both accumulation of autophagosomes and permeabilization/rupture of the lysosomal membrane (Fig. 8). It is likely that autophagy deficiency and lysosomal failure are occurring simultaneously. By inducing Hsp70.1 disorder, hydroxynonenal conceivably plays sinister roles in the occurrence of degeneration/death of Langerhans cells, leading to type 2 diabetes. 

Applying the data of the present monkey experimental paradigm to consider human diseases, both the dietary ω-6 fatty acid- (exogenous; from cooking oils, deep-fried foods, etc.) and the biomembrane phospholipid- (intrinsic; from the circumstance containing electromagnetic wave, air pollution, etc.) peroxidation product hydroxynonenal, when combined, conceivably plays crucial roles for the lysosomal cell death not only in the pancreas but also in the brain. The authors speculate that not only type 2 diabetes but also Alzheimer’s disease might occur in response to the exogenous and intrinsic hydroxynonenal by the long-standing lysosomal cell death in the corresponding organ [11].

# According to the helpful suggestion, Reference [72] was added to support an implication of hydroxynonenal to type 2 diabetes by the previous report.

The authors do hope these corrections are satisfactory for the final acceptance. 

With warmest regards, 

Tetsumori YAMASHIMA, MD, PhD 

Kanazawa University Graduate School of Medicine 

Japan

---

## [Decision Letter · Decision Letter 1]

25 Oct 2021

Hydroxynonenal causes Langerhans cell degeneration in the pancreas of Japanese macaque monkeys

PONE-D-20-37303R1

Dear Dr. Yamashima,

We’re pleased to inform you that your manuscript has been judged scientifically suitable for publication and will be formally accepted for publication once it meets all outstanding technical requirements.

Kind regards,

David Chau

Academic Editor

PLOS ONE

Additional Editor Comments (optional):

Reviewers' comments:

Reviewer's Responses to Questions

**Comments to the Author**

1. If the authors have adequately addressed your comments raised in a previous round of review and you feel that this manuscript is now acceptable for publication, you may indicate that here to bypass the “Comments to the Author” section, enter your conflict of interest statement in the “Confidential to Editor” section, and submit your "Accept" recommendation.

Reviewer #1: All comments have been addressed

Reviewer #2: All comments have been addressed

2. Is the manuscript technically sound, and do the data support the conclusions?

Reviewer #1: Yes

Reviewer #2: (No Response)

3. Has the statistical analysis been performed appropriately and rigorously? 

Reviewer #1: Yes

Reviewer #2: (No Response)

4. Have the authors made all data underlying the findings in their manuscript fully available?

Reviewer #1: Yes

Reviewer #2: (No Response)

5. Is the manuscript presented in an intelligible fashion and written in standard English?

Reviewer #1: Yes

Reviewer #2: (No Response)

6. Review Comments to the Author

Reviewer #1: (No Response)

Reviewer #2: (No Response)

7. PLOS authors have the option to publish the peer review history of their article (what does this mean?). If published, this will include your full peer review and any attached files.

Reviewer #1: No

Reviewer #2: No

---

## [Editor Report · Acceptance letter]

29 Oct 2021

PONE-D-20-37303R1 

Hydroxynonenal causes Langerhans cell degeneration in the pancreas of Japanese macaque monkeys 

Dear Dr. Yamashima:

I'm pleased to inform you that your manuscript has been deemed suitable for publication in PLOS ONE. Congratulations! Your manuscript is now with our production department. 

Kind regards, 

on behalf of

Dr. David Chau 

Academic Editor

PLOS ONE